# Efficient Classification of Long Documents via State-Space Models

**Peng Lu**[1,2,*] **Suyuchen Wang**[1,2,*], **Mehdi Rezagholizadeh**[1], **Bang Liu**[2], **Ivan Kobyzev**[1]

[1]Huawei Noah's Ark Lab, Canada

[2] Department of Computer Science and Operations Research, Université de Montréal

{peng.lu, suyuchen.wang, bang.liu}@umontreal.ca

{mehdi.rezagholizadeh, ivan.kobyzev}@huawei.com

## Abstract

Transformer-based models have achieved state-of-the-art performance on numerous NLP applications. However, long documents which are prevalent in real-world scenarios cannot be efficiently processed by transformers with the vanilla self-attention module due to their quadratic computation complexity and limited length extrapolation ability. Instead of tackling the computation difficulty for self-attention with sparse or hierarchical structures, in this paper, we investigate the use of State-Space Models (SSMs) for long document classification tasks. We conducted extensive experiments on six long document classification datasets, including binary, multi-class, and multi-label classification, comparing SSMs (with and without pre-training) to self-attention-based models. We also introduce the SSM-pooler model and demonstrate that it achieves comparable performance while being on average 36% more efficient. Additionally our method exhibits higher robustness to the input noise even in the extreme scenario of 40%.

## 1 Introduction

Since the emergence of large-scale pre-trained language models such as BERT (Devlin et al., 2019) and GPT3 (Brown et al., 2020), these transformer-based models have become popular solutions for many text classification and generation tasks. However, their benefit is constrained to short-length inputs when the computation resource is limited because attention module requires quadratic computation time and space. More specifically, each token in a sequence of length $N$ requires pairwise computation with all $N$ tokens, which results in $O(N^2)$ complexity. Such limitation makes transformer-based models hard to process long sequential data efficiently. There are many works aiming to improve the performance on Long Document Classi-

fication for transformers (Dai et al., 2022). One of the common approaches is to simply truncate long texts to a pre-defined length, e.g. 512, which makes pre-trained models to be applicable for them. Some work demonstrated this technique is not sufficient for long documents due to the missing of important information (Dai et al., 2022).

Another sort of technique attempts to reduce the computation overhead of attention-based systems. This problem has several relevant solutions, e.g. Sparse Attention models (Beltagy et al., 2020) and Hierarchical Attention models (Chalkidis et al., 2022). One of the important sparse attention methods is Longformer, which leverages local and global attention to reduce the computational complexity of the models and increases the input length up to 4096 tokens. Another popular sparse attention method is BigBird (Zaheer et al., 2020): besides the global and local attention, it includes extra random attention modules to attend to a pre-defined number of random tokens. Apart from designing sparse attention mechanisms, Hierarchical Transformers (HAN) like ToBERT (Pappagari et al., 2019) propose to construct systems on top of the conventional transformer (Chalkidis et al., 2022). Basically, the long text is first split into several chunks less than a fixed number, e.g. 200. Next, every chunk is encoded by a vanilla transformer one by one to form a collection of chunk representations and then another transformer processes the sequence of chuck representations.

Apart from the computation complexity issue, some works pointed out that the length extrapolation ability of self-attention-based models is quite limited (Press et al., 2022; Ruoss et al., 2023), namely, the transformer models perform poorly during inference if the sequence length of test data is beyond the sequence length of training data. As an order-invariant encoding mechanism, the self-attention-based encoder heavily relies on the Position Embedding (PEs) to model input orders, how-

---

*Research done during internship in Huawei Noah's Ark Lab (Montreal).

ever, these works demonstrate that the failure of transformers on long sequence is due to the limited length generalization ability of these position embedding methods. This also encourages exploring alternative architectures for the challenge of long document classification problems.

Recently, Gu et al. (2022) propose modeling sequential data with state-space models and show surprisingly good performance on a benchmark for comparing Transformers over long sequential data (Tay et al., 2021). However, this benchmark only consists of one single text classification task. It is still unclear about the efficacy and efficiency of state-space models for long document tasks compared to transformer-based models.

In this paper, we aim to fill this gap with a more comprehensive by conducting extensive experiments and analysis on six long document classification benchmarks. Besides, we develop an efficient long document system and show that the state-space-based models outperform self-attention-based models (including sparse attention and HAN) and yield consistent performance across datasets with much higher efficiency and more robustness to input noise.

## 2 Background and Methodology

### 2.1 State Space Model

In this section, we briefly introduce a recent long dependency modeling method utilizing State-space models to encode sequential data (Gu et al., 2022). The state-space model is defined by the following equations, which map the 1-dimensional continuous input signal $u(t)$ to an N-dimensional hidden state $x(t)$, and this hidden state is then projected to a 1-dimensional output $y(t)$:

$$x'(t) = \boldsymbol{A}x(t) + \boldsymbol{B}u(t),$$
$$y(t) = \boldsymbol{C}x(t) + \boldsymbol{D}u(t), \quad (1)$$

where $\boldsymbol{A}, \boldsymbol{B}, \boldsymbol{C}, \boldsymbol{D}$ are trainable parameters. A discrete sequence, like text, can be regarded as discretized data sampled from a continuous signal with a step size $\Delta$, and the corresponding SSM in the recurrence manner is:

$$h_k = \overline{\boldsymbol{A}}h_{k-1} + \overline{\boldsymbol{B}}x_k,$$
$$y_k = \overline{\boldsymbol{C}}h_k + \overline{\boldsymbol{D}}x_k,$$
$$\overline{\boldsymbol{A}} = (\boldsymbol{I} - \Delta/2 \cdot \boldsymbol{A})^{-1}(\boldsymbol{I} + \Delta/2 \cdot \boldsymbol{A}), \quad (2)$$

where $\overline{\boldsymbol{A}}$ is the discretized state matrix and $\overline{\boldsymbol{B}}, \overline{\boldsymbol{C}}, \overline{\boldsymbol{D}}$ share the similar formulas. The SSM can

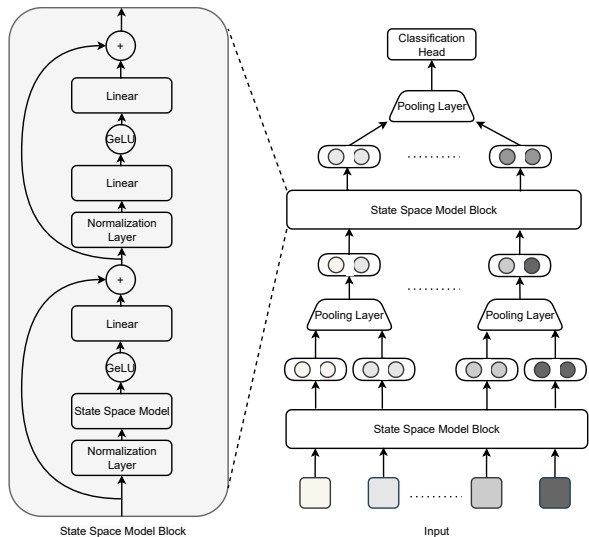

Figure 1: The illustration of the architecture of the State-Space-Pooler model.

also be rewritten in a linear convolution manner facilitating the encoding speed.

$$\overline{\boldsymbol{K}} = (\overline{\boldsymbol{C}\boldsymbol{B}}, \overline{\boldsymbol{C}\boldsymbol{A}\boldsymbol{B}}, ..., \overline{\boldsymbol{C}\boldsymbol{A}}^{L-1}\overline{\boldsymbol{B}}),$$
$$y_k = \sum_{j=0}^{k}(\overline{\boldsymbol{C}\boldsymbol{A}^j\boldsymbol{B}}) \cdot x_{k-j}, \quad (3)$$

where $\overline{\boldsymbol{K}} \in \mathbb{R}^L$ is defined as the SSM kernel. Given an input sequence $\boldsymbol{x} = \{x_1, \ldots, x_L\} \in \mathbb{R}^L$, the corresponding outputs $\boldsymbol{y} = \{y_1, \ldots, y_L\} \in \mathbb{R}^L$ of the convolution $\boldsymbol{y} = \overline{\boldsymbol{K}} * \boldsymbol{x}$ can be computed efficiently in $O(L \log L)$ time with the Fast Fourier Transform (FFT) (Cormen et al., 2009).

The computational complexity of the SSM algorithm is $O(L \log L)$ in convolution mode, which enables it to process significantly longer input sequences with the same hardware resources compared to the attention mechanism $O(L^2)$. This results in a substantial reduction in information loss from the input side. Next, instead of relying on a limited number of global attention mechanisms to capture long-term dependencies in sparse attention models, the SSM model is designed to model long sequences as dynamic systems. By utilizing a Hurwitz state matrix $A$ (Goel et al., 2022; Gu et al., 2023), it preserves long-term dependency information in a high-dimensional state. Such states enable SSM models strong ability to capture long dependency information.

Table 1: Test Performance on six LDU datasets with or without pre-training. We found SSM-based models achieve significant improvement over self-attention-based methods.

| Models | ECtHR | Hyperpartisan | 20News | EURLEX | BOOK | AMZ | Avg. |
|---|---|---|---|---|---|---|---|
| w/o pre-training | | | | | | | |
| **Transformer (6layer)** | 58.7 | 89.5 | 73.4 | 62.1 | 46.0 | 35.9 | 60.9 |
| **Longformer (6layer)** | 62.2 | 90.3 | **76.9** | 62.9 | 47.3 | 40.4 | 63.3 |
| **S4** | **64.4** | 93.8 | 76.2 | **72.6** | **49.7** | **44.9** | **66.9** |
| **S4-pooler** | 64.2 | **93.8** | 76.1 | 72.0 | 48.0 | 44.2 | 66.4 |
| w/ pre-training | | | | | | | |
| **BERT** | 71.7 | 91.8 | 84.7 | 73.2 | 58.2 | 51.1 | 71.8 |
| **BERT+random** | 72.8 | 89.3 | 85.0 | 73.3 | 59.2 | 56.8 | 72.7 |
| **BERT+textrank** | 73.5 | 91.2 | 84.7 | 72.9 | 58.9 | 56.9 | 73.0 |
| **Longformer** | 81.5 | 93.7 | 83.4 | 71.5 | 58.5 | 56.4 | 74.2 |
| **HAN** | 77.2 | 89.5 | 85.5 | 69.6 | 57.3 | 54.6 | 72.3 |
| **H3** | **82.9** | 94.0 | **85.9** | **76.7** | **60.9** | **57.9** | **76.4** |
| **H3-pooler** | 82.1 | **94.2** | 84.1 | 76.4 | 60.5 | 57.7 | 75.8 |

## 2.2 SSM-based Systems

In practice, recent SSM-based systems, e.g. S4 (Gu et al., 2022; Goel et al., 2022) are built up with a block as shown on the left of Figure 1. This block shares a similar structure as the self-attention module with pre-Layer Normalization (Ioffe and Szegedy, 2015; Ba et al., 2016) but replaces the multi-head attention with a discretized state-space model. The Structured State Space sequence model (S4) (Gu et al., 2022) utilizes this structure and leverages the Cauchy kernel method to simplify the kernel computation of SSM. Building on this idea, Hungry Hungry Hippo (H3) (Dao et al., 2022) extends the mechanism by including an additional gate and a short convolution obtained via a shift SSM to improve the language modeling ability of SSM-based systems.

## 2.3 State-Space-Pooler

To further improve the system efficiency, we propose a modification of the system architecture which imposes a progressive reduction of the input length for deeper layers. Figure 1 depicts our system architecture. The model progressively constructs the representation from the token level toward the final representation level. By inserting a max pooling layer between each SSM block, the model at each level automatically extracts the important information between nearby inputs and reduces the input length to half of the previous layer, which further accelerates the speed of training and inference. The final representation of the long sequence level is computed with the average of the last layer and then is fed to a fully-connected dense layer with softmax or sigmoid function to output the prediction probability for multi-class or multi-label problems, respectively.

## 3 Experiments

In this section, we present our experimental setup and results for multiple datasets.

### 3.1 Long Document Understanding (LDU)

**Dataset.** We first evaluated the effectiveness and efficiency on six commonly-used long text classification datasets including Book (Bamman and Smith, 2013), ECtHR (Chalkidis et al., 2021), Hyperpartisan (Kiesel et al., 2019), 20News (Lang, 1995), EURLEX (Chalkidis et al., 2019) and Amazon product reviews (AMZ) (He and McAuley, 2016). For the AMZ dataset, we randomly sampled product reviews longer than 2048 words from the Book category. We report accuracy for binary and multi-class classification tasks (Hyperpartisan, 20News, and AMZ) and macro-F1 for the rest of the multi-label classification problems. The detailed dataset statistics are included in Appendix A.

**Baselines.** We compare our methods with Transformer (Vaswani et al., 2017), Longformer and S4 with and without pre-training. More specifically, for models without pre-training, we compare 6-layer S4 and S4-pooler models (11m parameters) with 6-layer Transformer (70m parameters)

and Longformer models (99m parameters). For models with pre-trained checkpoints[1], we choose BERT-base with truncated input length and its two variants BERT-random and BERT-textrank (Park et al., 2022), we choose one sparse attention model: Longformer (Beltagy et al., 2020) and one hierarchical transformer model (HAN): ToBERT (Pappagari et al., 2019). The detailed model settings and hyperparameters are included in Appendix B.

**Results.** Table 1 shows the results of different methods on six commonly used long text classification datasets. Among the models without pre-training, the Transformer models perform more poorly than all other models by a large margin (3% - 6%). S4 and S4-pooler perform significantly better than Transformer and Longformer models on all datasets except for 20news, which is probably because the average length of this dataset is relatively short. On average, S4 and S4-pooler outperform the Longformer by 3.4% and 3.1%, respectively. For models fine-tuned with pre-trained checkpoints, although the BERT models with truncated inputs improve a lot compared to transformers without pre-training, they still underperform other methods, which indicates the necessity of processing longer text for this task. Again, the H3 and H3-pooler models give better performance on average. Besides, the S4-pooler and H3-pooler models demonstrate comparable performance to the corresponding original systems with around 36% training time reduction. We discuss the training time in the analysis section.

## 4 Discussion

In this section, we investigate the training efficiency, the impact of different down-sampling structures and the robustness to input noise of SSM-based models.

### 4.1 Time Analysis

Figure 2 shows the training time of different models on a single Nvidia V100 GPU for one epoch [2]. The maximum input length is set to 512 for transformer models and 4096 for Longformer, S4, and S4-pooler models. All models use the same batch size of 16 and layer number of 6. SSM-based models have a significant advantage in training effi-

[1]Our experiments were conducted with Huggingface toolkit (https://github.com/huggingface/transformers).
[2]The experiments of SSM models are adapted from (https://github.com/HazyResearch/safari).

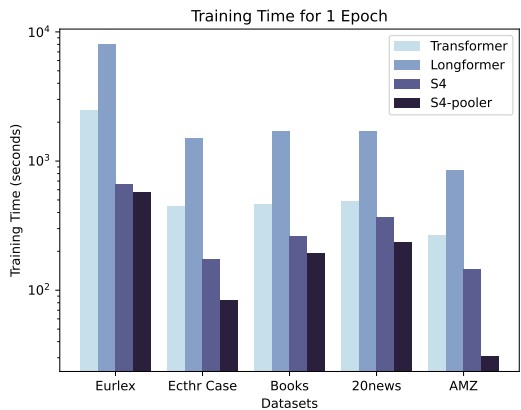

Figure 2: The training time comparison of 4 methods on five datasets. All models are trained on a single Nvidia V100 GPU for one epoch.

Table 2: Ablation study of different pooling structures.

| Models | AMZ | ECtHR | EURLEX | Book |
|---|---|---|---|---|
| S4 | 44.9 | 64.4 | 72.6 | 49.7 |
| S4-pooler-1-2 | 44.2 | 64.2 | 72.0 | 48.0 |
| S4-pooler-1-3 | 44.6 | 64.0 | 69.4 | 47.8 |
| S4-pooler-1-5 | 43.7 | 64.2 | 71.8 | 47.4 |
| S4-pooler-2-2 | 43.1 | 63.4 | 72.3 | 49.1 |
| S4-pooler-2-3 | 44.2 | 63.1 | 71.5 | 50.3 |
| S4-pooler-2-5 | 39.9 | 64.2 | 70.3 | 49.7 |

ciency compared to transformer and longformer models. Even with an 8 times greater maximum input length, S4 and S4-pooler models only took 42% and 27% of the training time of transformer, respectively. They also only took around 8% and 13% of the longformer's training time. By introducing pooling layers, the training time of the S4-pooler is reduced to 64% of S4 models on average.

### 4.2 Ablation Analysis

The pooling layer down-samples the input gradually, hence reducing the computation cost. Moreover, it changes the resolution of the input sequence by choosing different window sizes on different layers. We conducted experiments to investigate the impact of the hierarchical structures of S4-pooler models. S4-pooler-x-y refers to placing the pooling layer on every x layer with window size y. The results are averaged over three random seeds. The performance of different tasks is influenced by the hierarchical structures. For example, enlarging the down-sampling ratio from 2 to 5 decreases the performance of AMZ, while having less impact on the

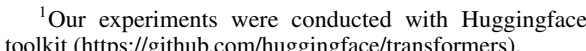

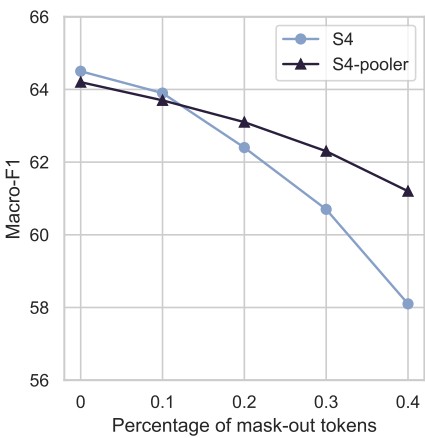

Figure 3: The performance of S4 and S4-pooler models on ECtHR datasets under different noise levels.

ECtHR task.

### 4.3 Robustness Analysis

Next, we analyze the robustness of SSMs to input noise. We conduct experiments on the ECtHR dataset under different input noise rates $\rho$. More specifically, we randomly mask out $\rho$ percent of inputs and train S4 and S4-pooler models with the noisy data. Figure 3 shows the results of two models. We can see the performance of S4 drops much faster than S4-pooler model with the increasing percentage of noise. This is reasonable because the S4-pooler system has max-pooling layers and imposes a stronger information extraction effect.

## 5   Conclusion

In this work, we conduct a comprehensive evaluation of the SSM-based models and show their superiority over self-attention-based models on long document classification tasks in terms of performance and training time. We further propose an efficient SSM-based system by imposing input length reduction for deeper layers and show our method performs on par with the original SSM models while greatly improving time efficiency.

## Limitations

The model we proposed only focuses on the classification of long documents, therefore, it may be extended to other NLP tasks, like long document Question Answering tasks. Furthermore, we concentrated on English datasets during the evaluation. In the future, we plan to extend it to more tasks and languages.

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

## A  Dataset

The dataset statistics is shown in Table 3. # class is the number of labels, the max, min, mean indicates the maximum, minimum and mean number of tokens with BERT tokenizer, respectively.

## B  Experimental setup details

We used Adam optimizer with default $\beta$ and we searched the best learning rate from 5e-5, 3e-5,

Table 3: Dataset statistics of LDU datasets.

| Dataset | # class | max | train | dev | test |
|---------|--------:|------:|-------:|------:|------:|
| ECtHR | 33 | 46,712 | 9,000 | 1,000 | 1,000 |
| Hyper | 2 | 5,538 | 516 | 64 | 65 |
| 20News | 20 | 30,602 | 10,182 | 1,132 | 7,532 |
| EURLEX | 4,721 | 4,443 | 45,000 | 6,000 | 6,000 |
| BOOK | 227 | 14,165 | 10,230 | 1,279 | 1,279 |
| AMZ | 5 | 18,282 | 3,880 | 485 | 485 |

1e-5 for one run of each baseline model and selected the best learning rate for the model. For the Transformer and Longformer trained from scratch, we used the BERT-base and Longformer-base and truncated them to be 6 layers without using the pretrained checkpoints. For all methods, we set the maximum epoch to 20 and selected the best model based on the performance metric on the dev set or the checkpoint of the last epoch. We reported the average results on the test set over three different seeds. For Experiments with S4, we followed the recommended setting (Gu et al., 2022). For H3 models, we used H3-125M which has a similar model scale as BERT-base and Longformer-base models.