# OpenReview forum: "Efficient Classification of Long Documents via State-Space Models"
_EMNLP/2023/Conference — EMNLP 2023 Main_

### Official Review · Reviewer_VUdY · 2023-07-31

**Soundness:** 4

**Excitement:**

3: Ambivalent: It has merits (e.g., it reports state-of-the-art results, the idea is nice), but there are key weaknesses (e.g., it describes incremental work), and it can significantly benefit from another round of revision. However, I won't object to accepting it if my co-reviewers champion it.

**Paper Topic And Main Contributions:**

This paper investigate State-Space Models for long document classification tasks. This paper conduct a comprehensive evaluation of the SSM-based models and show their superiority over self-attention-based models. The authors propose state space pooler to impose a progressive reduction of the input length for deeper layers.



**Questions For The Authors:**

- Why SSM-based models are more effective than self-attention-based models?

**Reasons To Accept:**

- The paper provide comprehensive evaluation of SSM-based models.  The experiments are conducted on self-attention-based models and SSM-based models on six commonly-used long text classification dataset with and without pre-training. Then the training efficiency and robustness analysis are also given.
- The proposed method is more efficient and robust than current methods.
- The authors shows the  superiority of SSM-based models over self-attention-based models.


**Reasons To Reject:**

- Their proposed method is simple pooling operation and lack of novelty.
- More in-depth analysis is lacked.

**Reproducibility:**

4: Could mostly reproduce the results, but there may be some variation because of sample variance or minor variations in their interpretation of the protocol or method.

**Reviewer Confidence:**

3: Pretty sure, but there's a chance I missed something. Although I have a good feel for this area in general, I did not carefully check the paper's details, e.g., the math, experimental design, or novelty.

---

> ### Author Rebuttal · Authors · 2023-08-25
>
> **R**: -   Why SSM-based models are more effective than self-attention-based models?
>
> >**A**: Due to the $O(N^2)$ time and space complexity, it is common to truncate the long sequential data for attention-based models, which is likely missing crucial information for addressing different tasks [1]. The complexity of SSM is $O(NlogN)$ in convolution mode, thus it is able to process much longer input given the same resources.
>
> >Apart from that, instead of modeling the long dependency with a limited number of global attention in the sparse attention, the SSM model is designed to model long sequences as a dynamical system, it preserves long dependency information in a high-dimensional state by utilizing a structured Hurwitz state matrix $A$. This enables SSM models to effectively capture long-term dependency information.
> >[1]Revisiting Transformer-based Models for Long Document Classification. Dai et al. EMNLP22.
>
> **R**: More in-depth analysis is lacked.
> >**A**: Thanks for your suggestion. We conducted more experiments to study the influence of different pooling strategies. All results will be included in the final version. The pooling layer changes the resolution of the input sequence via different window sizes. We conducted experiments to investigate the impact of the hierarchical structure of S4-pooler models. S4-pooler-x-y refers to placing the pooling layer on every x layer with window size y. The performance of different tasks is impacted by the resolution of the input sequence. For example, enlarging the down-sampling ratio from 2 to 5 decreases the performance of AMZ, while having less influence on the EcTHR task.
> |     **Models**    | **AMZ**  | **Ecthr**|**Eurlex**| **Book** |
> | :------------------- | :----------: | :----------: | :----------: | :----------: |
> | S4            |   44.9   |   64.4   |   72.6   |   49.7   |
> | S4-pooler-1-2 |   44.2   |   64.2   |   72.0   |   48.0   |
> | S4-pooler-1-3 |   44.6   |   64.0   |   69.4   |   47.8   |
> | S4-pooler-1-5 |   43.7   |   64.2   |   71.8   |   47.4   |
> | S4-pooler-2-2 |   43.1   |   63.4   |   72.3   |   49.1   |
> | S4-pooler-2-3 |   44.2   |   63.1   |   71.5   |   50.3   |
> | S4-pooler-2-5 |   39.9   |   64.2   |   70.3   |   49.7   |

---

### Official Review · Reviewer_FpWq · 2023-08-04

**Typos Grammar Style And Presentation Improvements:** L097
**Soundness:** 4

**Excitement:**

3: Ambivalent: It has merits (e.g., it reports state-of-the-art results, the idea is nice), but there are key weaknesses (e.g., it describes incremental work), and it can significantly benefit from another round of revision. However, I won't object to accepting it if my co-reviewers champion it.

**Missing References:**

1.  Jyun-Yu Jiang, Mingyang Zhang, Cheng Li, Michael Bendersky, Nadav Golbandi, and Marc Najork. 2019. Semantic
text matching for long-form documents. In Proceedings of the World Wide Web Conference. 795–806.
2. Jha, Akshita, et al. Supervised contrastive learning for interpretable long-form document matching. ACM Transactions on Knowledge Discovery from Data 17.2 (2023): 1-17.
3. Liu Yang, Mingyang Zhang, Cheng Li, Michael Bendersky, and Marc Najork. 2020. Beyond 512 tokens: Siamese multidepth transformer-based hierarchical encoder for long-form document matching. In Proceedings of the 29th ACM International Conference on Information and Knowledge Management. 1725–1734.

**Paper Topic And Main Contributions:**

This is a short paper that introduces State-Space Models for Long Document Classification Task. The authors conduct extensive experimentation on 6 long document classification task and demonstrate that state-space models outperform transformer-based self-attention models.

**Questions For The Authors:**

Did the authors experiment with pooling techniques for BERT-based models, i.e., instead of truncating to 512 tokens, would it help if the embeddings for different chunks were aggregated in some way to get a single embedding for a given long document for a fair comparison?


**Reasons To Accept:**

+ Conduct extensive experiments on 6 long document classification benchmark
+ Demonstrate that state-space models outperform transformer-based self-attention model
+ The proposed method is more efficient and more robust to input noises


**Reasons To Reject:**

- It would help to state the intuition on why state-space models perform better than transformer based models.
- The work might be considered slightly incremental.

**Reproducibility:**

3: Could reproduce the results with some difficulty. The settings of parameters are underspecified or subjectively determined; the training/evaluation data are not widely available.

**Reviewer Confidence:**

3: Pretty sure, but there's a chance I missed something. Although I have a good feel for this area in general, I did not carefully check the paper's details, e.g., the math, experimental design, or novelty.

---

> ### Author Rebuttal · Authors · 2023-08-25
>
> **R**:  It would help to state the intuition on why state-space models perform better than transformer based models.
>
> >**A**: Thanks for your suggestion, more discussion and missing references will be added in the final version.
>
> >Firstly, the computational complexity of the SSM algorithm is $O(NlogN)$ in convolution mode, which enables it to process significantly longer input sequences with the same hardware resources compared to the attention mechanism $O(N^2)$. This results in a substantial reduction in information loss from the input side.
>
> >Next, instead of relying on a limited number of global attention mechanisms to capture long-term dependencies in sparse attention models, the SSM model is designed to model long sequences as dynamic systems. By utilizing a Hurwitz state matrix A, it preserves long-term dependency information in a high-dimensional state. Such states enable SSM models strong ability to capture long dependency information.
>
> **R**: Did the authors experiment with pooling techniques for BERT-based models, i.e., instead of truncating to 512 tokens, would it help if the embeddings for different chunks were aggregated in some way to get a single embedding for a given long document for a fair comparison?
>
> >**A**: Thanks for your suggestion. We had such experimental comparisons and will make this part more clear in the final version. In Tab. 1, we compared our method to a Hierarchical Attention method (ToBERT) [1] for fair comparison. For this method, the long input is first split into several segments of less than 200 tokens. Next, each segment is encoded by a BERT model sequentially. The representation of the first token of each segment is then aggregated with a 2-layer transformer and the outputs of the transformer are fed to a mean pooling layer before the classification head.
>
> >[1] Hierarchical Transformers for Long Document Classification. Pappagari et al. IEEE ASRU 2019.

---

### Official Review · Reviewer_Y6PV · 2023-08-05

**Soundness:** 3

**Excitement:**

3: Ambivalent: It has merits (e.g., it reports state-of-the-art results, the idea is nice), but there are key weaknesses (e.g., it describes incremental work), and it can significantly benefit from another round of revision. However, I won't object to accepting it if my co-reviewers champion it.

**Paper Topic And Main Contributions:**

This paper proposes an empirical method that combines hierarchical pooling and the state space model.  This paper claims a contribution in that they first demonstrate the effectiveness of s4 on more text classification tasks. Another contribution it claims is the additional pooling layer, which improves time efficiency while harming performance.

**Questions For The Authors:**

1. I'd like to see more discussions about variations of the hierarchical mechanism, and the theoretical connection between it and s4 model.

**Reasons To Accept:**

1. This paper proposes a method to improve time efficiency over the state space model.

2. This paper evaluates the state space model on more datasets and investigates robustness.

**Reasons To Reject:**

1. It is not a surprise that the state space model works on more datasets.  And simply evaluating it on more datasets can not be a significant contribution.

2.  The proposed method improves efficiency at the cost of performance.

**Reproducibility:**

3: Could reproduce the results with some difficulty. The settings of parameters are underspecified or subjectively determined; the training/evaluation data are not widely available.

**Reviewer Confidence:**

4: Quite sure. I tried to check the important points carefully. It's unlikely, though conceivable, that I missed something that should affect my ratings.

---

> ### Author Rebuttal · Authors · 2023-08-25
>
> **R**:  It is not a surprise that the state space model works on more datasets. And simply evaluating it on more datasets can not be a significant contribution.
>
> >**A**:  The SSM was mainly evaluated on the Long Range Arena benchmark only consisting of one single text classification task. It is still unclear about the efficacy and efficiency of state-space models compared to attention-based models for long document tasks covering more domains and length varieties. Our work not only filled this gap but also provided a more efficient solution.
>
> **R**:  The proposed method improves efficiency at the cost of performance.
>
> >**A**: The training time reduction for long text classification is around 36% at the cost of only around 0.5% performance drop on average. Compared to the significant efficiency improvement, we think the performance drop is acceptable.
>
>  **Q**:  I'd like to see more discussions about variations of the hierarchical mechanism, and the theoretical connection between it and s4 model.
>
> > **A**: First, the pooling layer down-samples the input gradually, hence reducing the computation cost. Moreover, it changes the resolution of the input sequence by choosing different window sizes on different layers. We conducted experiments to investigate the impact of the hierarchical structures of S4-pooler models. S4-pooler-x-y refers to placing the pooling layer on every x layer with window size y. The results are averaged over three random seeds. The performance of different tasks is influenced by the hierarchical structures. For example, enlarging the down-sampling ratio from 2 to 5 decreases the performance of AMZ, while having less impact on the EcTHR task.
> |     **Models**    | **AMZ**  | **Ecthr**|**Eurlex**| **Book** |
> | :------------------- | :----------: | :----------: | :----------: | :----------: |
> | S4            |   44.9   |   64.4   |   72.6   |   49.7   |
> | S4-pooler-1-2 |   44.2   |   64.2   |   72.0   |   48.0   |
> | S4-pooler-1-3 |   44.6   |   64.0   |   69.4   |   47.8   |
> | S4-pooler-1-5 |   43.7   |   64.2   |   71.8   |   47.4   |
> | S4-pooler-2-2 |   43.1   |   63.4   |   72.3   |   49.1   |
> | S4-pooler-2-3 |   44.2   |   63.1   |   71.5   |   50.3   |
> | S4-pooler-2-5 |   39.9   |   64.2   |   70.3   |   49.7   |

---

### Meta-Review · Area_Chair_SNBG · 2023-09-18

**Recommendation:** 3

**Metareview:**

The paper extends State-Space Models to many longer benchmarks, and proposed a slightly adjusted way to make it more efficient. It's a clearly written paper. Its main merit is to further reassure the efficacy of SSMs.

pros

1. A solid justification for SSM to certain longer text documentation.

2. Well written and easy to follow.

cons

1. The proposed adjustment of model is not significant. None reviewer finds it novel enough such that the reviewer won't to champion it.

2. To make it a really comprehensive study-style paper on efficacy of SSMs on long document, the analysis is not enough and it lacks many other references to compare with.

---

### Decision · Program_Chairs · 2023-10-07

**Decision:**

Accept-Main

**Comment:**

The paper extends State-Space Models to many longer benchmarks, and proposed a slightly adjusted way to make it more efficient. It's a clearly written paper. Its main merit is to further reassure the efficacy of SSMs.

pros

1. A solid justification for SSM to certain longer text documentation.

2. Well written and easy to follow.

cons

1. The proposed adjustment of model is not significant. None reviewer finds it novel enough such that the reviewer won't to champion it.

2. To make it a really comprehensive study-style paper on efficacy of SSMs on long document, the analysis is not enough and it lacks many other references to compare with.